# Transduction Enhancers Enable Efficient Human Adenovirus Type 5-Mediated Gene Transfer into Human Multipotent Mesenchymal Stromal Cells

**DOI:** 10.3390/v13061136

**Published:** 2021-06-12

**Authors:** Robin Nilson, Olivia Lübbers, Linus Weiß, Karmveer Singh, Karin Scharffetter-Kochanek, Markus Rojewski, Hubert Schrezenmeier, Philip Helge Zeplin, Wolfgang Funk, Lea Krutzke, Stefan Kochanek, Astrid Kritzinger

**Affiliations:** 1Department of Gene Therapy, University Medical Center Ulm, 89081 Ulm, Germany; Robin.nilson@uni-ulm.de (R.N.); olivia.luebbers@hotmail.de (O.L.); linus.weiss@yahoo.de (L.W.); Lea.krutzke@uni-ulm.de (L.K.); astrid.kritzinger@uni-ulm.de (A.K.); 2Department of Dermatology and Allergology, University Medical Center Ulm, 89081 Ulm, Germany; karmveer.singh@uni-ulm.de (K.S.); karin.scharffetter-kochanek@uni-ulm.de (K.S.-K.); 3Institute for Transfusion Medicine, University Medical Center Ulm, 89081 Ulm, Germany; markus.rojewski@uni-ulm.de (M.R.); h.schrezenmeier@blutspende.de (H.S.); 4Institute for Clinical Transfusion Medicine and Immunogenetics Ulm, German Red Cross Blood Donation Service, 89081 Ulm, Germany; 5Schlosspark Klinik Ludwigsburg, Privatklinik für Plastische und Ästhetische Chirurgie, 71638 Ludwigsburg, Germany; p.zeplin@schlosspark-klinik.com; 6Schönheitsklinik Dr. Funk, 81739 München, Germany; info@schoenheitsklinik.com

**Keywords:** hMSC, mesenchymal stromal cells, mesenchymal stem cells, adenovirus, gene therapy, transduction enhancer, viral vectors, good manufacturing practice, GMP

## Abstract

Human multipotent mesenchymal stromal cells (hMSCs) are currently developed as cell therapeutics for different applications, including regenerative medicine, immune modulation, and cancer treatment. The biological properties of hMSCs can be further modulated by genetic engineering. Viral vectors based on human adenovirus type 5 (HAdV-5) belong to the most frequently used vector types for genetic modification of human cells in vitro and in vivo. However, due to a lack of the primary attachment receptor coxsackievirus and adenovirus receptor (CAR) in hMSCs, HAdV-5 vectors are currently not suitable for transduction of this cell type without capsid modification. Here we present several transduction enhancers that strongly enhance HAdV-5-mediated gene transfer into both bone marrow- and adipose tissue-derived hMSCs. Polybrene, poly-l-lysine, human lactoferrin, human blood coagulation factor X, spermine, and spermidine enabled high eGFP expression levels in hMSCs. Importantly, hMSCs treated with enhancers were not affected in their migration behavior, which is a key requisite for many therapeutic applications. Exemplary, strongly increased expression of tumor necrosis factor (TNF)-stimulated gene 6 (TSG-6) (a secreted model therapeutic protein) was achieved by enhancer-facilitated HAdV-5 transduction. Thus, enhancer-mediated HAdV-5 vector transduction is a valuable method for the engineering of hMSCs, which can be further exploited for the development of innovative hMSC therapeutics.

## 1. Introduction

Human multipotent mesenchymal stromal cells (hMSCs), a population of cells isolated from several adult tissues, are defined by the minimal criteria proposed by the ISCT in 2006 [1]. These criteria include expression or lack of specific surface markers, plastic adherent growth in vitro, and trilineage mesenchymal differentiation (into osteoblasts, adipocytes, and chondroblasts) [1]. In recent years, hMSCs have been identified as therapeutic candidates for regenerative medicine, tissue repair, oncolytic approaches, and other cell-based therapies [2,3,4]. They possess a range of unique characteristics that are inherently beneficial for various therapies and have been exploited in more than 10,000 clinical trials registered at www.clinicaltrials.gov (10 June 2021).

To further increase their therapeutic potential, hMSCs can be genetically modified. Among other viral and non-viral gene delivery strategies, transduction of cells by adenoviral vectors represents a safe and efficient method to modify target cells genetically [5,6]. Unlike the modification of cells with lentiviral vectors, adenoviral vector-based transduction does not harbor the risk of insertional mutagenesis since the adenoviral genome remains extrachromosomal. As adenoviral vectors have been studied in clinical trials for decades, they are well characterized, considered safe, and can be easily produced to high titers under good manufacturing practice (GMP) conditions [7].

A large proportion of the adenoviral vectors developed for gene therapy are based on adenovirus type 5 (HAdV-5). Internalization of HAdV-5 into cells occurs upon binding of the adenoviral penton base-associated RGD motif to the cell surface integrins α_v_β_3_ and α_v_β_5_ [8,9], proteins also present on the surface of hMSCs [10,11,12]. To enable efficient binding of the penton base to cell surface integrins in vitro, HAdV-5 particles need to attach to the cell surface via binding of the HAdV-5 fiber protein to the coxsackie and adenovirus receptor (CAR) [13]. As hMSCs largely lack CAR expression [10,12,14], transduction by HAdV-5-based vectors is severely limited. This bottleneck can be circumvented by genetic modification of the adenoviral vectors to enable CAR-independent transduction, e.g., by incorporating a RGD motif into the HAdV-5 fiber protein [10,15]. Several studies showed that utilization of specific reagents—herein referred to as transduction enhancers—poses an alternative to genetic modification of HAdV-5 vectors enabling efficient adenoviral transduction of CAR-negative cells with unmodified HAdV-5 vectors.

Several cationic polymers and lipids, including polybrene [16], poly-l-lysine [17] or the K2 transfection system [18], were reported to increase the HAdV-5 transduction efficiency in various cell types, including hMSCs [17,19,20]. Besides, some proteins are known to interact with the HAdV-5 capsid, thereupon mediating cell transduction, both in vivo and in vitro [21,22,23,24]. Examples are the human proteins lactoferrin and human blood coagulation factor X, both enabling CAR-independent cell transduction with HAdV-5 vectors [21,22,23,24].

To overcome the bottleneck of insufficient hMSC transduction with HAdV-5 vectors, we screened several compounds for their transduction enhancing effect. Here, we show that the known transduction enhancers polybrene, poly-l-lysine, human lactoferrin, and human factor X efficiently enhanced HAdV-5-mediated gene transfer into hMSCs. In addition, we identified the polyamines spermine and spermidine as even more efficient transduction enhancers. We determined the optimal amount of each transduction enhancer using HAdV-5 vectors expressing eGFP and showed that enhancers significantly improved transduction of both bone marrow-derived and adipose tissue-derived hMSCs. Beyond that, hMSC migration—a characteristic important for most therapeutic applications—was not negatively affected by enhancer-mediated HAdV-5 transduction in an in vitro Boyden chamber assay. Finally, we show that transduction enhancers can be used to enable efficient expression of therapeutic proteins in hMSCs, using TNF-stimulated gene 6 protein (TSG-6) as a model protein.

## 2. Materials and Methods

### 2.1. Human MSC Isolation and Cell Culture

Bone marrow (BM)- and adipose tissue (A)-derived human multipotent mesenchymal stromal cells (hMSCs) were manufactured and characterized by the Institute for Clinical Transfusion Medicine and Immunogenetics of the German Red Cross (Ulm, Germany) as described elsewhere [25,26]. Characterization of BM-hMSC included validation of the identity and purity following the ISCT minimal criteria, including expression (CD105, CD73, and CD90) and lack (CD45, CD34, CD3 and HLA-DQ, DP, DR) of specific surface markers, plastic-adherent growth, and trilineage differentiation [1]. A-hMSCs were analyzed for expression of CD13, CD105, CD73, and CD90 and lack of CD45, CD34, CD14, and HLA-DQ, DP, DR, based on the IFATS and ISCT joint statement [27]. Cells were expanded in BioWhittaker^®^ Alpha Minimum Essential Medium (Lonza Group AG, Basel, Switzerland) supplemented with 5% (A-hMSCs) or 8% (BM-hMSCs) irradiated pooled human platelet lysate (PL) (Institute for Clinical Transfusion Medicine and Immunogenetics [28]) and 500 units of heparin-sodium-5000 (Ratiopharm, Ulm, Germany).

UM-SCC-11B cells (a head and neck squamous cell carcinoma (HNSCC) cell line kindly provided by Prof. Cornelia Brunner, Clinic for Oto-Rhino-Laryngology, University Medical Center Ulm) were propagated in Dulbecco’s Modified Eagle’s Medium supplemented with 10% fetal bovine serum and 1× penicillin/streptavidin/glutamine (all Thermo Fisher Scientific, Waltham, MA, USA).

### 2.2. Adenoviral Vectors

All adenoviral vectors used in this study are *E1*-deleted and based on human adenovirus type 5 (HAdV-5) (GenBank ID: AY339865.1, sequence from nucleotide 1 to 440 and nucleotide 3523 to 35,935. Nucleotides 441 to 3522, comprising the *E1* gene region, are deleted. Adenoviral vectors expressing enhanced green fluorescent protein (eGFP) carried a CMV promoter-driven eGFP expression cassette, derived from pEGFP-N1 (Clontech 6085-1), instead of the deleted *E1*-region. The TSG-6-expressing vector had the eGFP sequence replaced with a cDNA coding for the human tumor necrosis factor (TNF)-stimulated gene 6 (TSG-6) protein (nucleotide accession GenBank: AJ419936.1, codon-optimized for mammalian gene expression by GeneArt (Thermo Fisher Scientific, Waltham, MA, USA)).

### 2.3. Adenoviral Vector Production and Purification

All HAdV-5 vectors were produced in *E1*-complementing N52.E6 cells [29]. Cells were transfected with linearized bacmid or plasmid encoding the adenoviral genome. After the initial rescue of viral particles, adenoviral vectors were amplified over several cycles to obtain a sufficient vector amount for final production and purification. For production, the amount of adenoviral vector was titrated to achieve a complete cytopathic effect after 48 h (corresponding to a physical multiplicity of infection (pMOI) of 300–500 (=300–500 viral particles per cell). Cells were harvested and resuspended in 50 mM HEPES buffer (pH 7.4) with 150 mM NaCl (Sigma-Aldrich, St. Louis, MO, USA) 48 h post-transduction. After freeze–thaw lysis of the cells, adenoviral particles were purified by a CsCl step gradient (1.41 g/mL CsCl and 1.27 g/mL CsCl; 2 h at 176,000× *g*, 4 °C) followed by a continuous CsCl gradient (1.34 g/mL; 20 h at 176,000× *g* and 4 °C). PD-10 size exclusion columns (GE Healthcare, Chicago, IL, USA) were used for buffer exchange before storing the particles at −80 °C (storage buffer: 50 mM HEPES, 150 mM NaCl, pH 7.4 with 10% glycerol). Total physical particle titers were determined by optical density measurement at 260 nm wavelength [30]. Quality control of all adenoviral vectors comprised restriction digest of isolated vector DNA, partial sequencing of the adenoviral genome, and SDS-PAGE with subsequent silver staining.

### 2.4. Transduction of hMSCs by HAdV-5-eGFP with or without Transduction Enhancers

3 × 10^4^ hMSCs were seeded into nunclon^®^ Δ 24-well plates (Thermo Fisher Scientific, Waltham, MA, USA). The next day, cells were washed with PBS, and 300 µL PL-free medium was added. Cells were transduced with HAdV-5-eGFP (pMOI 1000) after pre-incubation of viral particles with or without transduction enhancers, diluted in PL-free medium (total volume 60 µL, 30 min, 37 °C) As transduction enhancers, polybrene (Merck KGaA, Darmstadt, Germany), poly-l-lysine (0.01% solution; Sigma-Aldrich, St. Louis, MO, USA), human lactoferrin (Sigma-Aldrich, St. Louis, MO, USA), human blood coagulation factor X (Haematologic Technologies Inc., Essex Junction, VT, USA), spermine (Sigma-Aldrich, St. Louis, MO, USA), and spermidine (Sigma-Aldrich, St. Louis, MO, USA) were used. Three hours after transduction, cells were washed with PBS and further cultivated in PL-containing cultivation medium. Cells were harvested, and eGFP expression was analyzed using a CytoFLEX flow cytometer 72 h p.t. (Beckman Coulter, Brea, CA, USA).

### 2.5. Boyden Chamber Assays Using BM-hMSCs

2 × 10^5^ BM-hMSCs were seeded into nunclon^®^ Δ 6-well plates (Thermo Fisher Scientific, Waltham, MA, USA). The next day, cells were washed with PBS, and 1 mL PL-free medium was added. Cells were transduced with pMOI 1000 of the indicated HAdV-5 vector. Before transduction, viral particles were pre-incubated ± the indicated amount of enhancer for 30 min at 37 °C (300 µL total volume). Three hours after transduction, cells were washed with PBS, and PL-containing cultivation medium was added. After 2 h regeneration time cells were detached using TrypLE-select (Thermo Fisher Scientific, Waltham, MA, USA). 2 × 10^4^ cells were seeded into the upper chamber of tissue culture (TC)-inserts with a pore size of 8 µm (Sarstedt, Nümbrecht, Germany). TC-inserts were placed into 24-well plates harboring UM-SCC-11B cells to stimulate migration. The 1 × 10^5^ UM-SCC-11B cells were seeded in DMEM + 3% FBS 24 h before starting the migration assay. As a negative control, TC-inserts were placed into 24-well plates containing just DMEM + 3% FBS. Then, 18 h after seeding the transduced hMSCs, residual non-migrated BM-hMSCs were removed from the upper chamber via thoroughly cleaning with cotton swabs. Subsequently, migrated BM-hMSCs (at the bottom side of the TC-inserts) were fixed using ice-cold methanol for 15 min. Cell nuclei were stained with 4′,6-diamidino-2-phenylindole (DAPI; Sigma-Aldrich, St. Louis, MO, USA). Ten microscope images of different areas were taken per TC insert (10-fold magnification). Cell nuclei in these 10 images were counted, and the total number of migrated hMSCs per TC-insert was calculated.

### 2.6. Isolation of Macrophages (MΦ)

Human CD14^+^ peripheral blood mononuclear cell (PBMC) monocyte-derived macrophages were isolated from buffy coats (German Red Cross Blood Donation Service, Ulm, Germany) as described previously [31]. Isolation was performed by density centrifugation on Biocoll separating solution (1.077 g/mL; Biochrom, Cambridge, UK). PBMCs were resuspended in ammonium–chloride–potassium buffer to lyse residual red blood cells. CD14^+^ PBMCs were isolated using MACS magnetic beads (Miltenyi Biotec, Bergisch Gladbach, Germany). After washing the isolated CD14^+^ PBMCs with PBS, cells were seeded into Dulbecco’s Modified Eagle’s Medium (Sigma-Aldrich, St. Louis, Missouri, USA) supplemented with 10% fetal bovine serum (Biochrom, Cambridge, UK), 1× nonessential amino acid-mix (Sigma-Aldrich, St. Louis, MO, USA), 1 mM pyruvate (Sigma-Aldrich, St. Louis, MO, USA), 1× l-glutamine (Sigma-Aldrich, St. Louis, MO, USA), and 1× penicillin-streptomycin (Biochrom, Cambridge, UK). After 48 h, granulocyte-macrophage colony-stimulating factor (GM-CSF; Miltenyi Biotec, Bergisch Gladbach, Germany) was added to the medium to a final concentration of 0.01 µg/mL to induce differentiation of PBMCs to macrophages.

### 2.7. Co-Cultivation of BM-hMSCs with Activated Macrophages

Four days after starting differentiation of CD14^+^ PBMCs to MΦ, cells were detached using Accutase^®^ solution (Sigma-Aldrich, St. Louis, MO, USA). 2 × 10^5^ cells were added to the same amount of BM-hMSCs seeded into nunclon^®^ Δ 6-well plates (Thermo Fisher Scientific, Waltham, MA, USA) the day before. Simultaneously, macrophages were activated using lipopolysaccharide (LPS) (Sigma-Aldrich, St. Louis, MO, USA) and interferon-γ (INF-γ) (Sigma-Aldrich, St. Louis, MO, USA), which was added to the hMSC cultivation medium (10 ng/mL LPS and 20 ng/mL INF- γ) [31].

### 2.8. Transduction of BM-hMSCs with HAdV-5-TSG-6

hMSCs were seeded into 6-well plates as described previously. Then, 24 h later, BM-hMSCs were transduced with HAdV-5-TSG-6 (pMOI 1000) ± polybrene, spermidine, or factor X in PL-free medium. Three hours p.t., cells were washed with PBS and PL-containing medium was added to the cells.

### 2.9. Western Blot Analysis for Detection of TSG-6 in Cell Culture Supernatants

Seventy-two hours after starting the co-culture of BM-hMSCs with activated MΦ or after transduction with HAdV-5-TSG-6, 10 µL cell culture supernatant was used for sodium dodecyl sulfate–polyacrylamide gel electrophoresis (SDS-PAGE) with subsequent Western blot analysis. To detect TSG-6, a biotin-coupled goat α-TSG-6 polyclonal antibody (diluted 1:2000; R&D Systems, Inc., Minneapolis, MN, USA) and HRP-coupled streptavidin (diluted 1:4000; Agilent, Santa Clara, CA, USA) were used. The ImageQuant™ LAS 4000 system (GE Healthcare, Chicago, IL, USA) was used to detect chemiluminescence after adding the Luminata forte Western HRP-substrate (Merck, Darmstadt, Germany).

### 2.10. Statistical Analysis

Experiments performed in this study were repeated with hMSCs of three different healthy donors unless described otherwise. Statistical tests used for analysis are stated in the figure legends and were performed using GraphPad Prism software version 9.0.2 (GraphPad Software LLC, San Diego, CA, USA). A *p*-value ≤ 0.05 was considered as statistically significant.

## 3. Results

### 3.1. Polybrene, Poly-l-Lysine, Lactoferrin, Factor X, Spermine, and Spermidine Are Potent Enhancers of hMSC Transduction with HAdV-5 Vectors

#### 3.1.1. Determination of Optimal Transduction Enhancer Amounts

Due to the lack of CAR expression, hMSCs are hardly transduced by HAdV-5-based vectors (Figure A1 and [10,12,14]). To identify molecules enhancing the transduction of bone marrow-derived hMSCs (BM-hMSCs), different compounds and proteins were analyzed for their effect on HAdV-5-mediated transduction. While the cationic polymers polybrene and poly-l-lysine and the human proteins lactoferrin and factor X have previously been associated with transduction enhancement and binding to the adenoviral capsid, the polyamines spermine and spermidine, to our knowledge, have not been tested so far. We incubated eGFP-expressing HAdV-5 vectors with the different compounds before they were used to transduce BM-hMSCs. Different concentrations were tested to optimize transduction efficiency. Cells were analyzed by flow cytometry 72 h p.t. (Figure 1). Only mean fluorescence intensity (MFI) values of transduced hMSCs are shown since a precise determination of % eGFP-positive cells was not possible. We further discuss this issue in detail in the Appendix A (Figure A2).

All tested compounds increased transduction efficiencies in a dose-dependent manner. Optimal concentrations of each transduction enhancer were identified, and the respective enhancer-specific increase in eGFP expression compared to transduction without enhancer was calculated. The transduction enhancers polybrene (91-fold), poly-l-lysine (135-fold), and lactoferrin (100-fold) moderately improved transduction efficiencies, whereas factor X (235-fold), spermine (>1000-fold), and spermidine (>1000-fold) enabled very strong eGFP expression. When a certain threshold of the enhancer concentration was passed, a decrease in eGFP expression was observed for all enhancers. This effect was less pronounced for the proteins lactoferrin and factor X than for the other enhancers. Incubation of cells with high concentrations of spermine (≥1000 µg/mL) or spermidine (≥625 µg/mL) resulted in morphological changes and partial detachment of cells from the cell culture plates. The identified optimal amounts of enhancers per viral particle are listed in Table 1.

#### 3.1.2. Validation of Transduction Enhancer Activity in BM-hMSCs and A-hMSCs of Different Donors

BM-hMSCs of three healthy donors were transduced with HAdV-5 after pre-incubation with the determined optimal concentrations (Table 1) to investigate whether the transduction enhancing effect of the tested substances was valid for BM-hMSCs of different donors. Since adipose tissue-derived hMSCs (A-hMSCs) are also a frequent hMSC source, this cell type—also isolated from three different donors—was included in this study and treated in the same manner. Enhanced GFP expression was analyzed by flow cytometry 72 h p.t. (Figure 2a,b).

For both BM-hMSCs and A-hMSCs, transduction efficiencies were improved donor independently by all transduction enhancers. Similar to the previous experiments, factor X, spermine, and spermidine proved to be the most efficient adjuvants. The significantly improved transduction efficiency can also be seen in fluorescence microscopic images (Figure 2c).

### 3.2. BM-hMSC Migration Is Not Inhibited by HAdV-5 Transduction or the Transduction Enhancers

One of the reasons for the interest in hMSC-based therapies is the cell-autonomous ability of hMSCs to migrate to target sites (e.g., inflammatory sites, wounds, tumors). Using a Boyden chamber assay, we analyzed whether enhancer-facilitated transduction of hMSCs negatively affects their migration. BM-hMSCs were transduced with or without the addition of transduction enhancers. Subsequently, the cells were placed in Boyden chambers (8 µm pore size), and migration towards UM-SCC-11B-conditioned medium was analyzed after 18 h (Figure 3).

The migration assay revealed that hMSCs showed significantly increased migration towards the tumor cell-conditioned medium compared to the unconditioned medium. The number of hMSCs migrating to this stimulus was not decreased by adenoviral transduction, no matter if and which transduction enhancer was used. This held true for cells of all three donors tested in our experiments. Our data confirm that in vitro hMSC migration was not impaired by HAdV-5 vector transduction in the presence of transduction enhancers.

### 3.3. Transduction Enhancers Facilitate High-Level Expression of a Secreted Therapeutic Protein in BM-hMSC

To analyze whether BM-hMSCs can be engineered to express secreted therapeutic proteins from an adenoviral vector after enhancer-facilitated transduction, we performed transduction experiments with an adenoviral vector expressing TSG-6 (HAdV-5-TSG-6). TSG-6 is produced as a secreted protein in response to inflammatory mediators by many cell types [32,33,34]. Previously, hMSCs have been reported to secrete TSG-6 in response to cytokines released by activated macrophages (MΦ) [35]. We co-cultured BM-hMSCs of three donors with LPS and INF-γ-activated MΦ to stimulate endogenous TSG-6 expression. The amount of endogenously produced TSG-6 was then compared to the amount secreted by HAdV-5-TSG-6-transduced BM-hMSCs. For this purpose, culture supernatants from co-cultivation experiments or transduced hMSCs were collected after 72 h and analyzed for TSG-6 expression using Western blot analysis (Figure 4).

Co-cultivation of BM-hMSCs with LPS and IFN-γ-activated MΦ resulted in a significantly higher level of TSG-6 in the cell culture supernatant than in the supernatant of untreated BM-hMSCs (Figure 4a). This higher secretion resulted in an enhanced signal of TSG-6 protein itself (~35 kDa) and an enhanced quantity of inter-alpha-inhibitor-TSG-6 (IαI-TSG-6; ~120 kDa) complexes (discussed elsewhere [33,36,37,38]) in the supernatant. Semiquantitative densitometric analysis showed an ~8-fold increased TSG-6 signal compared to the supernatant of hMSCs co-cultured with non-activated MΦ. hMSCs transduced with HAdV-5-TSG-6 without enhancer showed no benefit compared to co-cultivation with activated MΦ regarding TSG-6 secretion. In contrast, polybrene, spermidine, or factor X as transduction enhancers significantly boosted TSG-6 secretion. The level of TSG-6 detected was so high that densitometric analysis was not feasible since the signal of hMSCs co-cultured with activated MΦ was only detectable when the signals derived from transduced hMSCs were already significantly oversaturated.

Nonetheless, Western blot analyses of the cell culture supernatants revealed a strongly improved secretion of TSG-6 after transduction of BM-hMSCs with HAdV-5-TSG-6 vector when combined with the enhancers polybrene, spermidine, or factor X. TSG-6 secretion following adenoviral transduction seems to be significantly higher than the endogenous expression of TSG-6 by cytokine-stimulated BM-hMSCs.

## 4. Discussion

Genetic modification of hMSCs may enhance their potential as cell therapeutics for many applications. Initially identified as having beneficial effects for several disorders, including bone repair [39], cardiovascular disease [40], or autoimmune disease [41], hMSCs are nowadays frequently modified genetically to improve clinical outcomes. For example, hMSCs were engineered to improve migration to, and retention at, the target site or enable efficient therapeutic protein production [42,43,44]. Moreover, the engineering of hMSCs enables previously inconceivable therapies, as hMSCs have been found to be promising carriers for oncolytic viruses [45]. Therein, hMSCs allow for virus replication and enable targeted transfer to tumor sites.

The modification of hMSCs with HAdV-5 vectors, the currently most frequently used vector type, has been of limited success due to the absence of the primary HAdV-5 receptor CAR on the hMSC membrane [10,12,14]. We tested several compounds for their potential to enhance HAdV-5-mediated transduction of hMSCs. All tested adjuvants were found to efficiently enhance transduction of both BM-hMSCs and A-hMSCs with HAdV-5 vectors. When transduced with an HAdV-5-eGFP vector (pMOI 1000), hMSCs hardly showed any eGFP expression (Figure 1 and Figure 2), indicating low transduction efficiency. However, when combined with polybrene, poly-l-lysine, or lactoferrin, eGFP expression was significantly enhanced. The transduction enhancers factor X, spermidine, and spermine turned out to enhance cellular transduction even further (up to 1000-fold). The transduction-enhancing effect of cationic polymers is in line with the results reported by other researchers. Not only polybrene [16] and poly-l-lysine [18,20], but also polyethylenimine (PEI) [46], the K2 transfection system [18], and lipofectamine [47,48,49] are cationic polymers and lipids known to enhance HAdV-5-mediated gene transfer in different cell types. As all these molecules share their cationic charge, it is hypothesized that these polymers and lipids form complexes with the negatively charged adenoviral particles, which can then efficiently enter the cells. As hMSCs lack CAR expression (Figure A1), this charge-associated mechanism probably enables CAR-independent adenoviral transduction of hMSCs.

The polyamines spermine and spermidine also have a positive charge and, to our knowledge, have previously not been reported to enhance cell transduction with HAdV-5 vectors. While a charge-mediated mode of action could explain the transduction enhancing effect, the impact of polyamines might go further. They are known to influence transcription, translation, and cell cycling and, therefore, play key roles in a wide variety of cellular processes [50]. Besides their importance in mammalian cells, polyamines also play essential roles in viral infection [51]. Spermine and spermidine, for example, were detected in the virions of herpes simplex virus (HSV-1) [52], are essential for ebolavirus (EBOV) gene expression and replication [53], and influence multiple stages in the life cycle of Coxsackievirus B3 (CVB3) including cellular attachment [54]. Polyamines appear to be very versatile biomolecules, and the stunning enhancing effect of spermine and spermidine on transduction-mediated transgene expression of hMSCs may have manifold reasons which might not only be ascribed to the charge of the molecule. The enhanced transgene expression might be influenced by additional factors other than solely an improved HAdV-5 vector uptake, which has to be investigated in further studies. In addition to the transduction enhancing effect of spermine and spermidine, we observed that incubation of cells with high concentrations of those molecules resulted in morphological changes and partial detachment of cells, indicating cytotoxic effects. This observation was not further investigated in this study; cytotoxic effects of the polyamines spermine and spermidine and their oxidation products have been noted and discussed before [55,56,57].

Lactoferrin and factor X have been reported to enable CAR-independent transduction of different cell types with HAdV-5, but, to our knowledge, have not been tested with respect to hMSCs. Johansson et al. [21] showed that human lactoferrin in tear fluid mediates enhanced HAdV-5 infection of CAR-negative epithelial cells. A similar mechanism has been described for human factor X binding to the HAdV-5 hexon protein and mediating binding of the viral particle to cell-surface heparan sulfate proteoglycans [24,58,59,60,61,62]. In our study, the uptake-enhancing effect of factor X was quite remarkable and resulted in a more than 200-fold enhancement of eGFP expression in hMSCs.

Besides achieving significant transgene expression levels in hMSCs, their intrinsic migration behavior should not be negatively affected by the transduction procedure for most applications. It is assumed that hMSCs migrate towards inflammatory sites and tumor tissue due to the local release of several cytokines and growth factors [63]. Vascular endothelial growth factor A (VEGF-A) [64], stromal cell-derived factor 1 (SDF-1) [65], and platelet-derived growth factor (PDGF) [66,67,68] are examples of such migratory attractants, the latter being assumed to be involved in the migration of hMSCs towards head and neck squamous cell carcinoma (HNSCC) [67]. Using a Boyden chamber assay to analyze the migration behavior of hMSCs towards an HNSCC cell line, we confirmed that enhancer-based transduction of hMSCs with HAdV-5 did not inhibit their migration in vitro, suggesting that the use of transduction enhancers represents a valuable resource for many therapies exploiting their inherent migratory properties (Figure 3).

To study the potential of HAdV-5 vector modified BM-hMSCs for the expression of secreted proteins, we chose the tumor necrosis factor (TNF)-stimulated gene 6 (TSG-6) as an example. TSG-6 is a multifunctional protein secreted by several cell types, including hMSCs [35], upon cytokine stimulation [33,36,69]. hMSCs have anti-inflammatory properties during wound healing [35,70,71] and in this process, TSG-6 has been shown to be an essential effector of hMSCs [35,72]. Qi et al. [28] reported that hMSCs suppressed the pro-inflammatory MΦ activation by LPS and INF-γ in vitro. This was likely caused by increased TSG-6 secretion from hMSCs stimulated with MΦ-released TNF-α [35]. Thus, adenoviral vector-mediated TSG-6 expression could benefit wound healing. Here, we compared TSG-6 protein levels in cell culture supernatants from hMSCs, either co-cultured with activated MΦ or transduced with HAdV-5-TSG-6 after pre-incubation with transduction enhancers (Figure 4). hMSCs co-cultured with activated MΦ resulted in an 8-fold increase in TSG-6 expression, similar to findings previously reported by Qi et al. [35]. Gene transfer with HAdV-5-TSG-6 alone resulted in no beneficial effect compared to co-cultivation of hMSCs with activated MΦ. However, when using polybrene, spermidine, and factor X for transduction enhancement, dramatically increased TSG-6 secretion was observed. 

In this study, we compared different transduction enhancers for their potential to improve HAdV-5-mediated gene transfer into hMSCs. Several compounds were identified, which dramatically improved transgene expression of both cellular and secreted proteins. As the inherent migratory behavior of hMSCs was preserved, the methods established here could pave the way for innovative therapies connecting the versatile and promising properties of hMSCs and adenoviral vectors. 

## Figures and Tables

**Figure 1 viruses-13-01136-f001:**
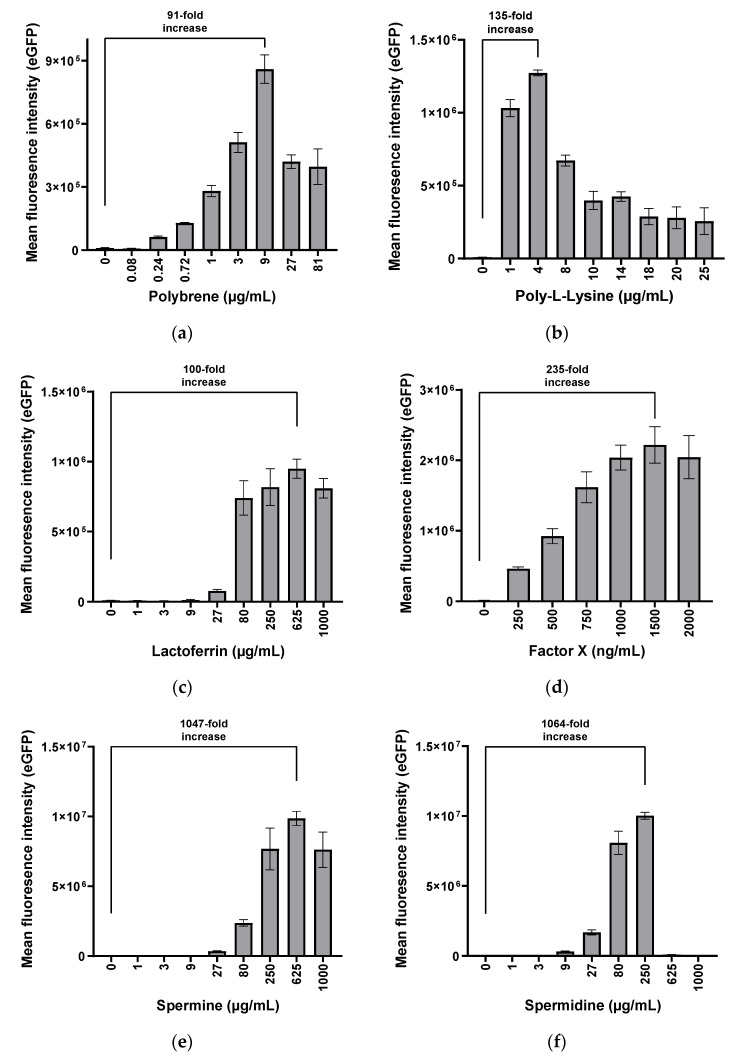
Identification and dose optimization of transduction enhancers to enhance transduction of BM-hMSCs with HAdV-5 vectors. BM-hMSCs were transduced with pMOI 1000 of an HAdV-5-eGFP vector. Before transduction, HAdV-5 vector particles were pre-incubated with polybrene (**a**), poly-l-lysine (**b**), human lactoferrin (**c**), human factor X (**d**), spermine (**e**), or spermidine (**f**) for 30 min at 37 °C. Transduction efficiency was analyzed by flow cytometry 72 h p.t. Results are shown as mean ± standard deviation of biological triplicates. A fold increase of mean fluorescence intensity relative to transduction without enhancer is provided for optimal enhancer concentrations.

**Figure 2 viruses-13-01136-f002:**
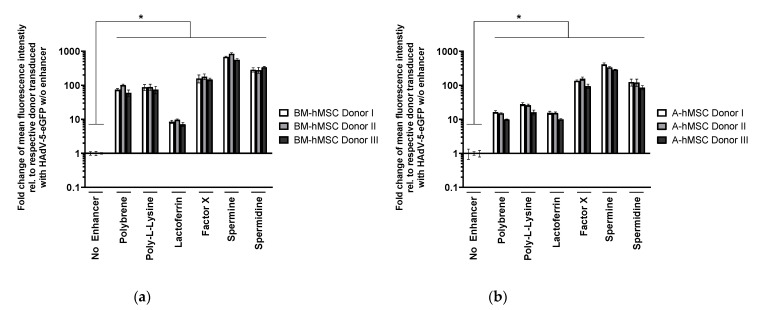
Use of transduction enhancers results in increased HAdV-5-mediated eGFP expression in BM-hMSCs and A- hMSCs of different donors. BM-hMSCs (**a**) and A-hMSCs (**b**) (of three healthy donors each) were transduced with HAdV-5-eGFP (pMOI 1000) after vector pre-incubation with the indicated enhancers at the optimal concentration (Table 1). Transduction efficiency was analyzed by flow cytometry 72 h p.t. Results are shown as mean ± standard deviation of biological triplicates. (**c**) Brightfield and fluorescence images of eGFP-expressing BM-hMSCs upon transduction with HAdV-5-eGFP (pMOI 1000) ± the indicated enhancer. For statistical analysis, an unpaired two-tailed student’s *t*-test was performed (* *p* ≤ 0.05).

**Figure 3 viruses-13-01136-f003:**
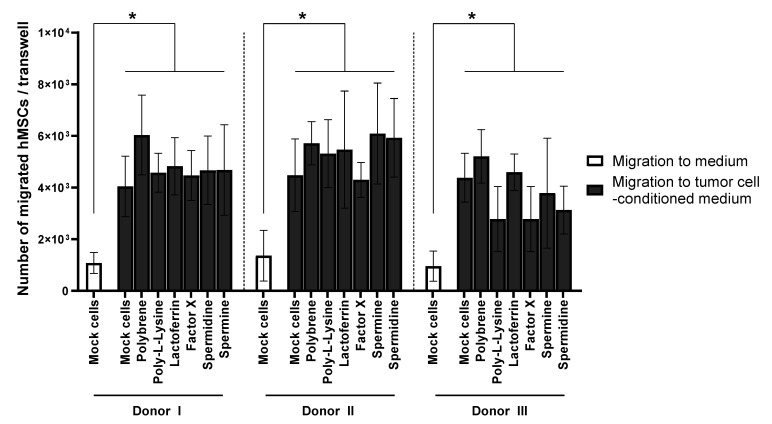
hMSC migration is not inhibited by HAdV-5 transduction in the presence of transduction enhancers. BM-hMSCs were transduced with HAdV-5-eGFP (pMOI 1000) pre-incubated with the indicated enhancers at optimal concentration (Table 1). After transduction, cells were detached and used in Boyden chamber assays. 2 × 10^4^ untreated (mock) or transduced BM-hMSCs were seeded into the upper chamber, while the lower chamber contained unconditioned or UM-SCC-11B-conditioned medium. After 18 h the total number of migrated hMSCs was determined. Results are given as mean ± standard deviation of biological triplicates. One-way ANOVA with subsequent Tukey’s multiple comparison was used for statistical analysis (* *p* ≤ 0.05).

**Figure 4 viruses-13-01136-f004:**
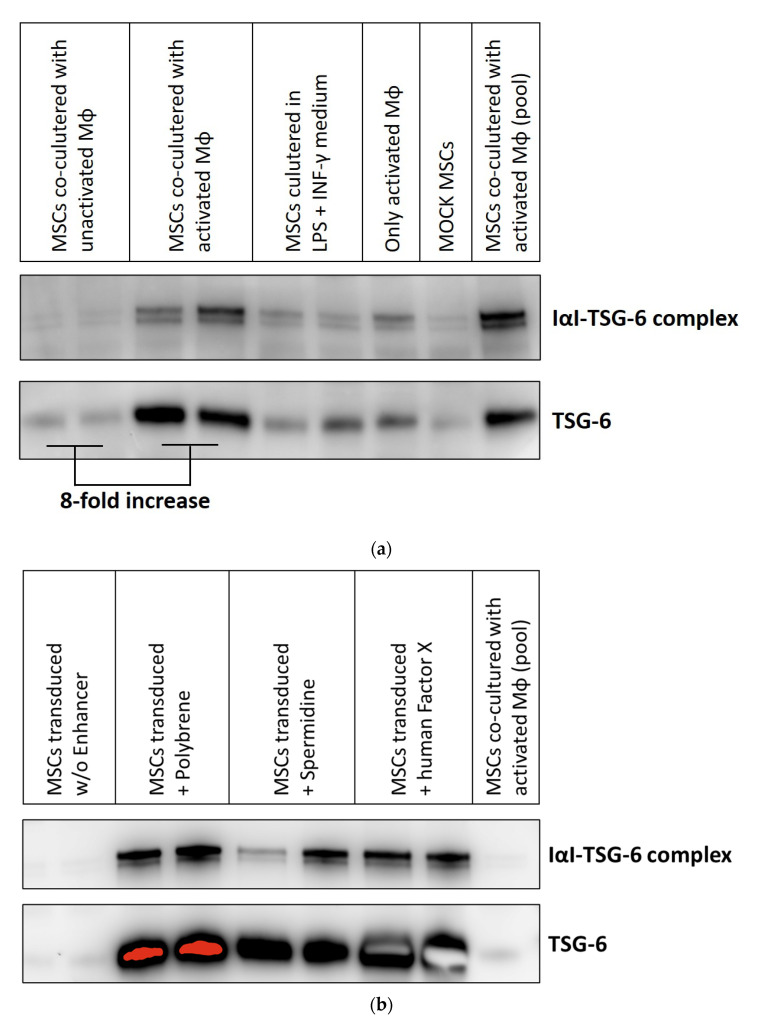
HAdV-5-mediated TSG-6 expression is strongly increased in BM-hMSCs using polybrene, factor X, or spermidine as transduction enhancers. Western blot analyses were performed using 10 µL of cell culture supernatant. For detection of TSG-6, a polyclonal goat α-TSG-6 antibody coupled to biotin was used in combination with HRP-coupled streptavidin. All conditions were analyzed in duplicates with cells from each of the three donors. Shown here are exemplary Western blot analyses of donor I. TSG-6 is detected as monomer and in complex with the inter-alpha-inhibitor (IαI-TSG-6). The duplicates of hMSCs co-cultured with activated MΦ were pooled and are shown in the last lanes of both blots to enable comparison of TSG-6 expression. (**a**) Western blot analysis of cell culture supernatants from hMSCs co-cultured with LPS and INF-γ-activated MΦ for 72 h. Supernatants of hMSCs co-cultured with non-activated MΦ or cultured in LPS and INF-γ-containing medium served as additional controls, apart from activated MΦ or untreated (mock) hMSCs alone. (**b**) Western blot analysis of cell culture supernatants from transduced BM-hMSCs 72 h p.t. Cells were transduced with HAdV-5-TSG-6 (pMOI 1000) ± the indicated enhancers.

**Table 1 viruses-13-01136-t001:** Optimal amount of transduction enhancers for BM-hMSC transduction.

Transduction Enhancer	Total Amount	Concentration during Pre-Incubation
Polybrene	18 fg/viral particle	9 µg/mL
Poly-l-Lysine	4% (*v*/*v*) *****	4% (*v*/*v*) *****
Human Lactoferrin	1250 fg/viral particle	625 µg/mL
Human Factor X	4 fg/viral particle	1500 ng/mL
Spermine	1250 fg/viral particle	625 µg/mL
Spermidine	500 fg/viral particle	250 µg/mL

* 4% *v*/*v* of a 0.01% poly-l-lysine solution was used.

## Data Availability

Data are contained and available within this manuscript.

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
