# Peer review of "Transduction Enhancers Enable Efficient Human Adenovirus Type 5-Mediated Gene Transfer into Human Multipotent Mesenchymal Stromal Cells"

_viruses, 2021, doi:10.3390/v13061136_

Round 1

Reviewer 1 Report

The manuscript by Nilson et al. describes the use of several different positively charged molecules to enhance transduction efficiency with adenovirus of cells that are normally resistant to infection due to low receptor expression. The authors use several previously described compounds, as well as some that were not used before to convincingly show that these various molecules enhance transduction of cells normally not infectable by adenovirus. They further show a practical application of these methods. Overall the paper is well presented with data supportive of the conclusions. The language is clear and the paper is easy to read. I think the topic would be useful to many. I think the paper is suitable for publication pending some minor revisions as outlined below.

  1. Line 52. I’m not sure “episomal” is the correct phrasing here as that generally refers to plasmids or circular DNA molecules, adenovirus genome remains linear so perhaps “extrachromosomal” would be more accurate?
  2. Line 87. “Exemplary” is not the correct word here, it means “outstanding”, I think what the authors mean here is an example. Same for figure 2c legend.
  3. Line 111. It is not clear what the E1-deleted vectors is, is it dl309-based or some other variant? This needs to be clarified.
  4. Materials and methods section 2.6. Since human blood was used, how was it obtained? What procedures and ethics guidelines were followed with human subjects? This needs to be added.
  5. Figure 1. It was odd to see spermidine drop so dramatically in enhancement of transduction going from 250 to 625 ug/ml, were intermediated concentrations tried? Any plausible explanation? Was cell viability affected by the high concentrations?
  6. Figure 2. What was the total percentage of cells that were transduced with and without the adjuvants?
  7. Figure 2 and 4. How long were the transgenes expressed after transduction? Was this investigated? Would be nice to know.
  8. Line 334. Would be nice to have the quantification in the figure.
  9. Line 356. I think it should say "Therein", this sentence should be rephrased as it makes it sound like it was shown in this study.
  10. Figure A1. I would include this as the first figure in the manuscript.

Reviewer 2 Report

This manuscript entitled ‚ Transduction enhancers enable efficient human adenovirus type 5-mediated gene transfer into human multipotent mesenchymal stromal cells’ by Robin Nilson et. al. invested six different transduction enhancers for their potential to improve HAdV-5-mediated gene transfer into hMSCs.

The author showed ~100-fold enhancement in hMSCs transduction with four previously reported molecular Polybrene, poly-l-lysine, human lactoferrin and human blood coagulation factor X. Moreover, two additional molecular spermine and spermidine increased the transduction further to 1000-fold compared to without.

This significant transduction enhancement in hMSCs is amazing; it makes hMSC cell therapeutics with Ad5 possible. In general, the study is well designed; the examination was detailed and complete. However, some issues should be clarified or discussed before it goes to public:

  1. The whole study is based on hMSCs ex-vivo transduction. In the initial comparison using Ad5-GFP, the fold change was only expressed by Mean fluoresence intensity; however, the percent of GFP positive cells is also important information. This should be showed for both figure 1 and fig2.
  2. The next question is, for hMSCs ex-vivo transduction, AAV or lenti-vector were used, too. How is the advantage/ difference of choosing adenovirus? Can the author compare the transduction efficiency in positive cells percentage (as in the last question)?
  3. A ‘pMOI 1000’ was applied in transduction of BM-hMSCs with both HAdV-5-eGFP and HAdV-5-TSG-6. Although in the Materials and Methods part (2.3 Adenoviral vector production and purification) the pMOI was descript, it is not clear. For example, how many viral particles equal to A ‘pMOI 1000’? What a viral particles/transducing unit ratio was calculated?
  4. As it is known, these transduction enhancers function via forming complexes with the negatively charged adenoviral particles to efficiently enter the cells. Then this effect is non cell-type specific. How would be the in vivo outcome?
  5. In Figure 1F, with 625 and 1000 μg/mL Spermidine no number is recorded/measured. Is this cytotoxic effect or other reason?

Round 2

Reviewer 2 Report

please include your answer and figures from comment 1 and 5 to the result and discussion part of the paper, they are meaningful Information to the authors.
